# The Roles of the Let-7 Family of MicroRNAs in the Regulation of Cancer Stemness

**DOI:** 10.3390/cells10092415

**Published:** 2021-09-14

**Authors:** Yuxi Ma, Na Shen, Max S. Wicha, Ming Luo

**Affiliations:** 1Department of Internal Medicine, Division of Hematology and Oncology, University of Michigan, Ann Arbor, MI 48109, USA; yuxima@hust.edu.cn (Y.M.); nashen@hust.edu.cn (N.S.); 2Cancer Center, Union Hospital, Tongji Medical College, Huazhong University of Science and Technology, Wuhan 430022, China; 3Department of Breast and Thyroid Surgery, Union Hospital, Tongji Medical College, Huazhong University of Science and Technology, Wuhan 430022, China

**Keywords:** cancer stem cells (CSCs), tumor suppressor microRNAs, long non-coding RNAs (lncRNAs), circular RNAs (circRNAs), lin28, glucose metabolism

## Abstract

Cancer has long been viewed as a disease of normal development gone awry. Cancer stem-like cells (CSCs), also termed as tumor-initiating cells (TICs), are increasingly recognized as a critical tumor cell population that drives not only tumorigenesis but also cancer progression, treatment resistance and metastatic relapse. The let-7 family of microRNAs (miRNAs), first identified in C. elegans but functionally conserved from worms to human, constitutes an important class of regulators for diverse cellular functions ranging from cell proliferation, differentiation and pluripotency to cancer development and progression. Here, we review the current state of knowledge regarding the roles of let-7 miRNAs in regulating cancer stemness. We outline several key RNA-binding proteins, long non-coding RNAs (lncRNAs) and circular RNAs (circRNAs) involved in the regulation of let-7 biogenesis, maturation and function. We then highlight key gene targets and signaling pathways that are regulated or mutually regulated by the let-7 family of miRNAs to modulate CSC characteristics in various types of cancer. We also summarize the existing evidence indicating distinct metabolic pathways regulated by the let-7 miRNAs to impact CSC self-renewal, differentiation and treatment resistance. Lastly, we review current preclinical studies and discuss the clinical implications for developing let-7-based replacement strategies as potential cancer therapeutics that can be delivered through different platforms to target CSCs and reduce/overcome treatment resistance when applied alone or in combination with current chemo/radiation or molecularly targeted therapies. By specifically targeting CSCs, these strategies have the potential to significantly improve the efficacy of cancer therapies.

## 1. Introduction

MicroRNAs (miRNAs) are a small class of non-coding RNAs identified as evolutionary conserved, single-strand RNAs with the size of 18–25 nucleotides. Functionally, miRNAs repress target gene expression by binding to the 3${}^{\prime }$-untranslated regions (3${}^{\prime }$UTRs) of target messenger RNAs (mRNAs) to promote degradation and prevent their translation into protein products. Since the initial discovery of C. elegans miRNA gene (lin-14) in the early 1990s [1], over 2000 miRNAs have been identified, which regulate diverse cellular processes, including proliferation, differentiation, survival, stem cell maintenance and tumorigenesis [2,3]. It has been well established that miRNAs play crucial roles in tumor development and progression by regulating key cancer regulatory genes, including oncogenes and/or tumor suppressor genes [2].

Let-7 (lethal-7) was initially identified as a heterochronic gene in *C. elegans* (the second miRNA gene identified after lin-14), where its expression determines adult cell fate in the worm [4].The let-7 family of miRNAs, which is comprised of multiple paralog genes located in different chromosomes, has been defined as one of the largest and most conserved family of miRNAs across different species, ranging from worms to humans [5,6,7]. In humans, there are 10 mature let-7 miRNAs, including let-7a, let-7b, let-7c, let-7d, let-7e, let-7f, let-7g, let-7i, miR-98 and miR-202, which are derived from 13 precursor genes [8].

During mammalian embryonic development, let-7 expression is silent in embryonic stem cells (ESCs) but reactivated at later development stages [5]. This role of let-7 in ESC differentiation is further validated by the findings that exogenous expression of let-7 in miRNA-deficient ESCs rescues their defective differentiation [9]. In cancer cells, the let-7 family of miRNAs generally function as tumor suppressor miRNAs to inhibit tumor growth and metastasis [8,10,11]. Let-7 miRNAs are also involved in suppressing CSC characteristics, including sphere/colony formation, tumor growth, differentiation and regenerative potential [8,10,11].

Cancer stem cells (CSCs), or tumor-initiating cells (TICs), represent a small subset of cells in the tumor that maintain or acquire the capacity to indefinitely self-renew as well as to differentiate into various tumor cell progenies constituting the bulk of a tumor mass [12,13]. As CSCs are endowed with the intrinsic capacity for therapeutic resistance, they contribute not only to tumorigenesis but also to disease progression and metastatic relapse. As a result of these properties, CSCs play a major role in the resistance of tumors to the conventional chemotherapy and radiation therapy [13].

miRNAs, especially the let-7 family of miRNAs and their regulatory proteins, are frequently deregulated in a wide variety of cancer types and influence CSC maintenance, metabolism, tumorigenesis and metastasis. This review summarizes the current state of knowledge on the roles of let-7 miRNAs and their regulatory proteins in regulating CSCs during tumor development and cancer progression, the signaling pathways and network through which let-7 miRNAs are involved in the regulation of CSCs. We also emphasize existing strategies that have been documented in the literature to harness this family of miRNAs for therapeutic benefit in preclinical models.

## 2. Let-7 as Key Tumor Suppressor miRNAs and Negative Regulators of CSCs

The CSC model posits that cancer arises from a rare population of tumor-initiating cells (TICs) or CSCs that possess dysregulated self-renewal capacity. Yu et al. presented the first evidence that let-7 miRNAs are markedly reduced in TICs but increased in differentiated tumor cells in breast cancer [14]. This increased let-7 expression in differentiated tumor cells (or non-TICs) inhibits the expression of H-Ras and HMGA2, known let-7 target genes that promote tumor self-renewal and proliferative capacity. In contrast, antagonizing let-7 by antisense oligonucleotides enhances self-renewal capacity of non-TICs, indicating a critical role of let-7 miRNAs in suppressing tumorigenicity by downregulating CSC properties [14].

In addition to a role in regulating breast CSCs (BCSCs), let-7 miRNAs are also implicated in CSC regulation in other tissue malignancies. For instance, let-7a is downregulated in stem-like cells in head and neck cancers as evidenced by their expression of aldehyde dehydrogenase (ALDH). Furthermore, increased let-7a expression represses chemoresistance and tumourigenicity through the ablation of CSC-like properties [15].

In human pancreatic cancer cells, CD44+/CD133+/EpCAM+ CSC-like cells exhibited lower levels of let-7 expression [16], and this suppressed let-7 expression promoted the expression of pluripotency transcription factor, such as NANOG, SOX2, SOX9, OCT4, KLF4 and c-myc, to maintain CSCs. In gastric cancer, let-7b was shown to directly suppress c-myc expression to promote gastric stem cell differentiation and drug sensitivity [17].

In urothelial carcinoma of the bladder or UCB, the activation of pro-inflammatory COX2/PGE2 signaling induces promoter methylation of let-7, resulting in its downregulation and subsequent upregulation of SOX2, which promotes the maintenance of urothelial CSCs [18]. In therapy-induced neuroendocrine prostate cancer (t-NEPC), upregulated Lin28B suppresses let-7 miRNA expression, which, in turn, promotes the de-repression of HMGA2 and HMGA2-mediated SOX2 expression, leading to increased CD44+ and CD133+ CSC-like cell populations [19].

Epithelial-mesenchymal transition or EMT, a cellular process facilitating tumor cell invasion, migration and dissemination has been shown to associate with the acquisition of mesenchymal CSC-like properties [20]. Increased Lin28A expression remarkably increases E-cadherin but reduces Vimentin expression and, subsequently, increases the epithelial CSC marker ALDH activity and colony formation via the downregulation of let-7a in breast cancer [21].

The suppression of let-7d expression activates key EMT transcriptional factors, including Twist and Snail, and promotes chemo-resistant abilities of oral squamous cell carcinoma [22]. In tongue squamous cell carcinoma (TSCC), the long noncoding RNA (lncRNA) H19, by binding to (sponging) let-7a to inhibit its function in repressing let-7 target gene HMGA2, facilitates TSCC cell migration and invasion [23].

In advanced high grade serous ovarian cancer (HGSOC), knockdown of Snail upregulates let-7 expression and reverses the cancer stemness phenotype [24]. Similarly, in HGSOC, low levels of let-7 expression were associated with increased EMT and higher spheroid-forming and tumor growth capacities [25]. The overexpression of mir-98, a member of let-7 family of miRNAs, reverses EMT through directed targeting of HMGA2 and transcriptional repression of the EMT-related gene POSTN [26].

## 3. Regulatory Mechanisms of Let-7 Expression and Function

### 3.1. Let-7 Biogenesis

The biogenesis of let-7 is similar to that of other miRNAs. Canonical let-7 miRNAs are first transcribed from the corresponding miRNA transcription units by RNA polymerase II to produce primary miRNAs (pri-miRNAs), which are subsequently processed by a complex containing an RNase III-like enzyme Drosha and its co-factor DGCR8, a double-stranded RNA-binding protein (RBP), into a 60–70nt-long precursor miRNAs (pre-miRNAs) in the nucleus [27,28,29,30].

The pre-miRNAs are then transported to the cytoplasm by exportin5 and further cleaved by Dicer, a cytoplasmic RNase III, to generate 18–25 bp long miRNA duplex [31,32,33,34,35]. One of the miRNA strands (called guide strand) in the duplex recruits Argonaute proteins (AGO1–4) to form RNA-induced silencing complex (RISC), while the other strand (passenger strand) is degraded. The RISC mediates post-transcriptional repression of miRNA (i.e., let-7) target genes, where the guide strand of miRNAs acts as a tether to the 3′UTR of specific mRNAs to cause their degradation and translational repression [36].

### 3.2. Post-Transcriptional Regulation of Let-7 by RNA-Binding/Processing Proteins

Let-7 constitutes a 13-member tumor suppressor miRNA family downregulated in a large fraction of tumor types [6,7]. Let-7 suppresses the translation of a number of oncogenes, including Myc, K-Ras and HMGA2 [37,38,39]. The biogenesis of miRNAs, including let-7, is regulated at different stages in the nucleus and cytoplasm by miRNA processing proteins, such as Drosha and Dicer, respectively, and shRNA-mediated downregulation of Drosha or Dicer1-enhanced tumor development in a K-Ras-induced mouse model of lung cancer, suggesting that the abrogation of global miRNA processing promotes tumorigenesis [40].

Subsequently, by examining the consequence of heterozygous vs. homozygous deletion of Dicer1 in mouse models, Kumar et al. demonstrated that the loss of a single allele of Dicer1 enhanced tumorigenesis, while the loss of both alleles had the opposite effect [41]. Indeed, reduced Dicer expression has been shown to predict poor patient survival in a number of cancer types, including chronic lymphocytic leukemia [42], breast [43], lung [44], ovarian [45], colorectal [46] and bladder [47,48] cancer.

In colorectal and breast cancer, low levels or the impairment of Dicer expression promotes epithelial-to-mesenchymal transition (EMT), metastasis formation and the acquisition of cancer stemness [49,50,51]. In parallel with the role of Dicer functioning as a haplo-insufficient tumor suppressor, recent studies also identified Dicer as a HIF-1α-interacting protein to promote CSC and metastatic activities in multiple types of cancer cell lines and human tumors, where HIF-1α suppresses Dicer expression by promoting its ubiquitination and autophagy-mediated degradation, leading to repressed maturation of known tumor suppressor miRNAs, including let-7 and miRNA-200b [52].

In contrast to Dicer1, DGCR8 and AGO RBPs that mediate let-7 (and other miRNAs) maturation and subsequent silencing of target mRNAs, a subset of RBPs can compete or inhibit miRNA-mediated mRNA silencing by binding sequences at or in close proximity to miRNA recognition elements (MREs) [53]. Of these, Lin28A and Lin28B (collectively referred as Lin28) are best known to suppress let-7 expression and function and to influence stem cell maintenance, metabolism and tumorigenesis [54,55,56,57,58].

Lin28A and Lin28B, two highly related RBPs and proto-oncogenes, inhibit let-7 biogenesis through distinct mechanisms by binding the let-7 precursor (pre-let-7) to interfere with the miRNA-processing-machinery-mediated cleavage [59,60,61]. Lin28B binds to pre-let-7 and prevents its processing by the miRNA-processing machinery in the nucleus, whereas Lin 28A functions in the cytoplasm to recruit terminal uridylyl transferase (TUT4 or TUT7), which places a short oligo (U) stretch to the 3${}^{\prime }$-end of pre-let-7, which blocks its processing by Dicer, leading to pre-let-7 degradation [61,62,63].

Lin28A and Lin28B are aberrantly expressed in a plethora of tissue malignancies, and activation of either Lin28A or Lin28B is responsible for the global post-transcriptional downregulation of let-7 miRNAs in many cancers, which is associated with advanced tumor stages and poor patient outcomes [55]. Through gain-or loss-of-function studies in various cancer cell lines and mouse models, Lin28A/B has been shown to promote tumor growth, invasion and metastasis [19,57,64,65].

In addition to Lin28, other RBPs, such as the insulin-like growth factor 2 mRNA-binding proteins (IMPs), have been characterized to compete with miRNAs for binding to MREs on target mRNAs and protect these transcripts from miRNA-mediated degradation [66]. In glioblastoma, IMP2 prevents let-7 target gene silencing by binding to let-7 MREs, thereby protecting them from degradation, leading to the preservation of glioblastoma stem-like cells [67]. In addition to IMP2, other IMP paralogs, such as IMP1 and IMP3, have also been shown to protect let-7 target genes from silencing [66]. Therefore, the IMPs fulfill a function similar to Lin28 and may cooperate with Lin28 to induce or maintain stemness in tumor cells.

### 3.3. Post-Transcriptional Regulation of Let-7 by LncRNAs and circRNAs

As depicted in Figure 1, a growing number of studies have shown that long non-coding RNAs (lncRNAs) and circular RNAs (circRNAs) play important roles in tumorigenesis and progression via the regulation of miRNAs, including the let-7 family. The human H19 encodes a lncRNA that acts as a trans regulator of the imprinted gene network controlling growth in mice [68]. H19 harbors both canonical and non-canonical binding sites for the let-7 family of miRNAs, which acts as a molecular sponge for let-7 [69].

Emerging evidence indicated that H19 plays important roles in regulating tumorigenicity and cancer stemness. For example, high levels of H19 are enriched in breast cancer stem cells (BCSCs), where it functions as a competing endogenous RNA to sponge let-7, leading to increased expression of the core pluripotency factor Lin28, thereby, promoting BCSC maintenance [70]. Through its sponging function, H19 also facilitates the migration and invasion of tongue squamous cell carcinoma cells by suppressing let-7a, leading to the activation of its target gene HMGA2, which, in turn, promotes EMT [23].

In esophageal cancer cells, elevated H19 expression promotes EMT and metastasis through activation of the STAT3/EZH2/β-catenin axis, and, interestingly, H19 was directly targeted by let-7c, suggesting a mutual negative-feedback regulation between H19 and let-7c [71]. Like H19, the long intergenic non-coding RNA or Linc-ROR, a regulator of cell reprogramming, plays a key role in accelerating the self-renewal capacity of CSCs in pancreas cancer by sponging some members of the let-7 family [72].

In addition to serving as a molecular sponge by binding to let-7, some LncRNAs regulate the let-7 family of miRNAs indirectly by modulating the expression of specific proteins that suppress let-7 maturation or function. In colorectal cancer, lncRNA PVT1-214 is markedly increased, which represses let-7 miRNA expression indirectly by inhibiting the degradation of Lin28A, which, in turn, drives the acquisition of stem cell-like properties [73].

In addition to lncRNAs, circular RNAs (circRNAs) are a type of single-stranded RNAs that, unlike linear RNA, form a covalently closed continuous loop. Some circRNAs contain MREs for the let-7 family of miRNAs. For example, circHMCU modulates breast cancer cell proliferation and mesenchymal characteristics by sponging let-7 miRNAs and, subsequently, activates let-7 targeted genes, including HMGA2, c-myc and CCND1 [74].

In non-small cell lung cancer (NSCLC) cells, let-7 is inhibited by circ-CPA4 via its molecular sponge function, and the knockdown of circ-CPA4 inhibited cell growth, mobility and promoted cell death by downregulating the release of exosomal PD-L1, which promotes CSC-like activities and resistance of NSCLC cells to cisplatin [75].

## 4. CSC Regulatory Pathways Involving Let-7 miRNAs

### 4.1. Wnt/β-Catenin Signaling

Aberrant Wnt/β-catenin signaling promotes CSC self-renewal and proliferative capacity, thus, exerting crucial roles in tumorigenesis and therapy response. Wnt/β-catenin signaling is activated in CSCs of multiple types of cancer. As shown in Figure 2, Wnt signaling molecules, including Wnt1, β-catenin, TCF and CCND1 have been reported to be regulated by the let-7 family of miRNAs. Several studies have indicated that let-7 miRNAs are involved in controlling breast cancer stemness through the mutual regulation of Wnt signaling.

For example, the Wnt/β-catenin pathway represses let-7 expression post-transcriptionally by the transactivation of Lin28B, a well-characterized suppressor of let-7 miRNA biogenesis, to augment breast CSC phenotypes [76]. In ER+ breast cancer, elevated let-7c expression decreases the tumorigenicity of CSCs and enhances the antitumor efficacy of tamoxifen through suppression of the ER and the Wnt pathway [77,78]. By acting as a specific molecular sponge of let-7c, lncRNA H19 restrains let-7c availability, promoting the symmetric division and maintenance of breast CSCs via the activation of ER-Wnt signaling [79].

In contrast to H19, miR-146a, via indirect upregulation of let-7c, promotes asymmetric division and inhibits the self-renewal ability of breast CSCs by downregulating Wnt signaling [80]. Through inhibition of the cyclin D1/Akt1/Wnt1 signaling pathway, let-7d also inhibits CSC properties and exerts synergistic effects with radiotherapy to inhibit breast CSC self-renewal [81].

A similar role of let-7b in controlling the self-renewal of breast CSCs was also suggested by a study showing that Matrine, a natural product extracted from the root of Sophora flavescens Ait, induced cisplatin hypersensitivity by suppressing the expression of Lin28A, resulting in the inactivation of Wnt pathway in a let-7b-dependent manner [82].

Paralleling its effects on breast cancer cells, Matrine was also reported to have antitumor properties in other types of cancer through upregulation of let-7b, which inactivates the Wnt/CCND1 signaling pathway and inhibits EMT and lung CSC activities [83]. In esophageal cancer, Wnt activation stimulates the self-renewal of CSC-like cells, which is blocked by the administration of nano-let-7b via direct inhibition of TCF-4/β-catenin complex activity to sensitize chemotherapeutic response [84].

In hepatocellular carcinoma cells, the overexpression of let-7a also inhibits the tumorsphere-forming ability by inhibition of EMT factors and the Wnt signaling pathway [85]. In NSCLC, oncogenic miR-367 promotes the self-renewal ability of CSC-like cells by degrading E3-Ligase Enzyme F-box and WD repeat domain-containing 7 (FBXW7) and repressing let-7c, leading to the activation of Wnt signaling [86]. Attenuation of Wnt/β-catenin activity by the tankyrase inhibitor JW74 leads to the induction of let-7 miRNA, which contributes to osteosarcoma cell differentiation [87].

### 4.2. NOTCH/Hedgehog Signaling

Aberrant activation of NOTCH signaling is associated with the stemness phenotype of many different tissue malignancies. Let-7 miRNAs play a functional role in the development of human neural progenitors by suppressing HMGA2 and NOTCH to regulates gliogenesis [88]. Recent studies have suggested mutual regulation of miRNAs and NOTCH signaling in multiple tumor cell types (Figure 2).

For example, treatment of pancreatic cancer cells with metformin, an antidiabetic drug, decreased the expression of the CSC markers CD44, EpCAM, EZH2, NOTCH1, NANOG and OCT4 but elevated the expression of miRNAs, including let-7a, let-7b, miR-26a, miR-101, miR-200b, and miR-200c in a mouse xenograft model [89]. Conversely, overexpression of NOTCH1 increased the expression of oncogenic miR-21 while decreasing the expression of miR-200b, miR-200c, let-7a, let-7b and let-7c, promoting EMT and a CSC-like phenotype in pancreatic cancer cells [90].

In ER+ breast cancer, miR-129 inhibits estrogen receptor 1 (ESR1)-mediated NOTCH signaling in CSC-like cells by suppressing cyclin D1/DICER1 mediated let-7 maturation, leading to increased NUMB expression to repress NOTCH signaling activity [91].

The hedgehog pathway is involved in embryonic development and maintenance of CSCs through signal transmission via sonic hedgehog (Shh), PTCH1, SMO to activate transcription factor GLI-driven downstream gene responses. As illustrated in Figure 2, activation of the hedgehog pathway can suppress the expression of let-7 miRNAs, which promotes CSC properties.

In pancreatic cancer, highly metastatic cancer cells exhibit upregulation of Shh and CSC surface marker expression, including CD133 and CXCR4, associated with lower levels of let-7 expression as compared to parental BxPC-3 cells [92]. In NSCLC, a hedgehog inhibitor promotes sensitization of tumor cells to erlotinib by increasing let-7c but suppressing the expression of CSC-associated genes, including SOX2, NANOG and EpCAM [93].

### 4.3. STAT3/NFκB/Cytokines Signaling

The activation of transcription factors, such as the signal transducer and activator of transcription 3 (STAT3) and NFκB, elicited by pro-inflammatory cytokines, such as IL1, IL6 and IL8, secreted in the tumor microenvironment, promotes tumor epigenetic/phenotypic plasticity reflected by increased mesenchymal signature gene expression and cancer stemness. As illustrated in Figure 3, one of the key mechanisms by which STAT3/NFκB signaling promotes cancer stemness is through their negative regulation of let-7 miRNAs.

In NSCLC, the oncogenic MUC1-C transmembrane protein induces EMT by promoting NFκB p65 binding to Lin28B chromatin to activate Lin28B expression, leading to the suppression of let-7 [94]. As is the case for NFκB, the activation of STAT3 in breast cancer promotes the transcription of Lin28B by directly binding to the Lin28B promoter, resulting in the repression of let-7 expression and concomitant upregulation of the let-7 target, HMGA2 [95].

Similarly, activation of the STAT3/NFκB pathway in breast cancer cells by M1 macrophage secreted pro-inflammatory cytokines elicits Lin28B-let-7-HMGA2 axis to induce CD44+/CD24- and ALDH1+ CSCs through activation of EMT [96]. In esophageal cancer cells, elevated let-7c expression following knocking down of lncRNA H19 is reported to repress cell proliferation, migration, invasion as well as EMT and metastasis via inhibition of the STAT3-EZH2-β-catenin pathway [71].

Overexpression of let-7c, which acts as an upstream suppressor of the Ras/NFκB signal pathway, decreases malignant transformation and acquisition of cancer stem cell-like properties in normal adult skin keratinocytes [97]. In oral cancer cells, ectopic expression of let-7c downregulates stemness and radio- and chemo-resistance through directly targeting IL8 [98]. In breast cancer cells, let-7a directly binds to the 3′UTR of the C-C chemokine receptor type 7 (CCR7) to suppress its expression, leading to impaired migration and invasion [99].

Let-7a is also reported to directly inhibit IL6 expression and activation of Src oncoprotein promotes NFκB-Lin28 signaling to inhibit let-7a expression, leading to high level of IL6 production and activation of NFκB, thereby, completing a positive feedback loop to facilitate an epigenetic switch of immortalized breast epithelial cells into a stably transformed CSC-like state [100].

### 4.4. MAPK/ERK and PI3K/AKT Signaling

Receptor tyrosine kinases (RTKs), such as the epidermal growth factor receptor (EGFR) family, are activated by extracellular ligands such as EGF to activate MAPK/ERK (also known as Ras-RAF-MEK-ERK) and PI3K-AKT-mTOR signaling pathways to regulate cell proliferation/self-renewal, survival and differentiation. The Ras small GTPases consist of three major isoforms, including H-, K- and N-Ras. Both H- and K-Ras are targeted by the let-7 family of miRNAs to regulate CSC activities (Figure 3).

Specifically, Lin28A facilitates glioblastoma neurosphere formation through constitutively activating K-Ras and suppressing let-7b and let-7g [101]. In pancreatic cancer, K-Ras blocks let-7i maturation by phosphorylating Lin28B at S243 and promoting Lin28B nuclear translocation [102]. The dietary agents sulforaphane, quercetin and catechins inhibit CSC-like activities of pancreatic cancer cells through let-7a induction and K-Ras inhibition [103]. Inhibition of the RAS/MEK pathway decreases the expression of let-7 target gene HMGA2, which is required to maintain EMT and the mesenchymal phenotype [104].

In K-Ras-driven lung cancer, increased Lin28B expression induces CD44+/CD326+ CSC-like cells by decreasing let-7 miRNAs and promoting AKT activity [105]. In Table 1, we listed most of the target genes involved in many different signaling cascades that are directly or indirectly repressed by let-7 family of miRNAs in various cancer types.

## 5. The Let-7/Lin28 Axis in Regulation of Stem Cell Metabolism

Earlier studies using transgenic mouse models demonstrated a role for the let-7/Lin28 axis in regulation of glucose metabolism. Transgenic mice expressing Lin28A have increased body size, crown-rump length and delayed onset of puberty due to increased glucose metabolism and insulin sensitivity [111]. This increased glucose metabolism is attributed to Lin28A/B-mediated repression of let-7, leading to an insulin-sensitized state through activation of the insulin-PI3K-mTOR pathway [111].

Such a role of the let-7/Lin28 axis in regulation of glucose metabolism is further demonstrated in transgenic mice with muscle-specific loss of Lin28A or overexpression of let-7, which results in insulin resistance, hyperglycemia and impaired glucose tolerance [112]. In addition to a role in regulating glucose metabolism during development, recent studies indicated that Lin28A and its closely related paralog, Lin28B, play critical roles in regulating pluripotent stem cells (PSCs).

Lin28, along with OCT4, SOX2, and NANOG, acts as important tissue factors to reprogram human somatic fibroblasts into pluripotent stem cells [113], which exist in a highly oxidative “naïve” state and a more glycolytic “primed” state [114,115]. Overexpression of Lin28A and Lin28B both enhance the derivation efficiency of induced PSCs (iPSCs), while loss of endogenous Lin28A/B reduces reprogramming efficiency and traps the derived mouse iPSCs in a more “naïve” oxidative state, suggesting that Lin28A and Lin28B facilitate transition from naïve to primed pluripotency [116].

As reflected by their multipotent tumor cell state, CSCs exhibit a remarkable ability to adapt their metabolism to survive and proliferate under adverse environment conditions, such as hypoxia, acidosis, and nutrient starvation. As illustrated in Figure 4, recent studies have provided strong evidence that the let-7/Lin28 axis regulates cancer progression and CSC activity through metabolic modulation.

In breast cancer, Lin28A and Lin28B enhance, whereas let-7 suppresses, aerobic glycolysis through regulation of pyruvate dehydrogenase kinase 1, or PDK1, in a hypoxia- or hypoxia-inducible factor-1 (HIF-1)-independent manner, suggesting direct regulation of the Lin28/let-7 axis in the modulation of glycolytic metabolism even at ambient oxygen levels [106]. PDK1, by phosphorylating pyruvate dehydrogenase (PDH) to inhibit its enzymatic function of promoting glucose flux to mitochondrial oxidative phosphorylation (OXPHOS), serves as a gatekeeper of aerobic glycolysis.

Interestingly, PDK1 is reported to be enriched in ALDH+ CSCs, and the depletion of PDK1 diminishes the sphere-forming ability and tumor growth by abrogating ALDH+ CSCs [117]. This increased PDK1 activity in BCSCs is linked to elevated expression of LncRNA H19, which acts as a molecular sponge sequestering let-7 miRNA, leading to the activation of HIF1α to enhance its downstream target PDK1 expression [117].

The expression of carbonic anhydrase IX (CAIX), a pH regulator and hypoxia marker, is correlated with poor prognosis and increased metastases in breast cancer [110]. In CAIX-suppressed breast cancer cells, let-7 miRNAs are upregulated while Lin28 protein levels decreased, leading to suppressed PDK1 activity and attenuated phosphorylation of PDH at Ser-232, which results in increased OXPHOS but reduced glycolysis and lactate production [110]. Thus, modulation of the Lin28/let-7 axis regulates tumor glycolytic metabolism through direct or indirect regulation of PDK1 activity, which directs glucose influx into aerobic glycolysis.

As mentioned above, overexpression of Lin28 and/or downregulation of let-7 miRNAs has emerged as an oncogenic driver in many different cancers, which promotes cancer stemness and poor patient outcome. The oncometabolite polyamine modulates the expression of eukaryotic translation initiation factor eIF-5A, which serves as a direct regulator of the Lin28/let-7 axis [108]. In neuroblastoma cells, Difluoromethylornithine (DFMO), a drug targeting ornithine decarboxylase, the rate-limiting enzyme of polyamine biosynthesis, reduces Lin28B protein expression while increasing let-7 miRNA, leading to the inhibition of glycolytic metabolism and decreased neurosphere forming activity [109].

In addition to glucose metabolism, cancer cells also rely on de novo fatty acid biosynthesis to maintain rapid cell proliferation and tumor growth [118]. Elevated expression of Lin28A/B enhances de novo fatty acid synthesis to promote cancer progression via SREBP-1 [107]. By direct association with the mRNAs of both SREBP-1 and SCAP, Lin28A/B enhances the translation and maturation of SREBP-1, which, in turn, accelerates metabolic conversion of saturated and unsaturated fatty acids and protects cancer cells from lipotoxicity [107].

The inhibition of Lin28B impairs cell growth and amino acid metabolism in acute myeloid leukemia by de-repressing let-7a, although the downstream molecular mechanisms are not known [119]. Thus, targeting Lin28 and/or restoring let-7 expression, by inhibiting tumor metabolic activities in CSCs, may provide potential therapeutic strategies for cancer treatment.

## 6. Let-7 miRNAs as Potential Therapeutic Agents Targeting CSCs

As mentioned above, the let-7 family of miRNAs have been extensively documented to serve as a class of tumor suppressor miRNAs to inhibit stem cell function in normal and cancerous tissues, and the enhanced expression of let-7 inhibits the maintenance of CSCs, thus, inhibiting tumor growth and progression. In fact, let-7 miRNA replacement therapy has now emerged as a powerful strategy to inhibit tumor growth and progression and overcome treatment resistance.

Three approaches to harness tumor suppressor miRNAs, such as let-7, for therapeutic purposes have been described, including: (1) the utilization of recombinant lentiviruses or adenoviruses expressing let-7 miRNAs in tumor cells as a type of gene therapy, (2) the administration of small molecules that epigenetically enhances endogenous let-7 expression, and (3) the administration of synthetic let-7 miRNA mimics in conjugation with nanoparticle-based delivery.

### 6.1. Let-7 Replacement Therapy Employing Plasmid and Virus-Based Expression Systems

To restore let-7 miRNA expression, a plethora of studies have used plasmid and virus-based vector systems to either directly express let-7 or express regulatory proteins capable of upregulating let-7 miRNA expression in tumor cells. The expression of let-7g is extremely low in hepatocellular carcinoma (HCC) and re-expression of let-7g miRNA in HCC cells inhibited tumor cell proliferation and migration via inhibition of K-Ras/HMGA2/Snail axis [120].

Moreover, the intra-tumoral injection of plasmid expressing let-7g inhibited tumor growth of HCC cells in nude mice [120]. Stable expression of let-7a in human breast cancer cells delivered by lentivirus infection significantly reduced mammosphere formation and tumorigenicity when implanted in immunodeficient mice [14]. Similarly, in ER+ breast cancer cells, lentivirus-mediated let-7c expression has been shown to decrease tumorigenicity of estrogen-treated BCSCs in vivo by suppression of Wnt signaling [78]. Over-expression of CDX2, a transcriptional factor called caudal type homeobox 2 in MCF7 breast cancer cells alleviates tumor growth and micro-metastasis by up-regulating let-7b and inhibiting COL11A1 expression [121].

Although overexpression of let-7 miRNAs in cancer cells has been shown to inhibit tumorigenicity and progression when implanted in mice, several studies have applied different virus-based systems for the direct delivery of let-7 miRNAs locally or intratumorally. For example, intranasal administration of recombinant adenovirus expressing let-7a RNA hairpin has been shown to significantly reduce tumor formation in vivo in the lungs of animals that express a G12D activating mutation for the K-Ras oncogene [122].

The ectopic expression of let-7g in K-Ras(G12D)-expressing murine lung cancer cells induced cell cycle arrest and cell death and overexpression of let-7g by inducible lentiviral vectors led to significant growth reduction in tumor xenografts [123]. A single dose of intratumor injection of lentiviruses expressing let-7c is capable of suppressing the tumor growth of prostate cancer xenografts [124].

### 6.2. Restoration of Endogenous Let-7 Expression to Enhance Treatment Responses

Current frontline therapies frequently fail to eradicate cancer due to their failure to eliminate cancer-stem-like cells, which confer resistance to traditional chemo/radiation or molecularly targeted therapies, promoting metastasis and cancer relapse. As shown in Table 2, therapy-induced cancer stemness by widely used chemotherapeutic or molecularly targeted agents has been reported to be abolished by restoration of let-7 miRNA expression in various types of cancer, including breast cancer [82], NSCLC [83], head and neck cancer [15], ovarian cancer [24,25] and others. In pancreatic cancer, Gemcitabine-resistant tumor cells exhibit higher CSC marker CD133 expression associated with lower levels of Let-7 [92,125].

Combined use of gemcitabine with thiostrepton, a natural cyclic oligopeptide antibiotic, synergistically inhibits sphere formation of NSCLC by elevating miR-98 [126]. In ER+ breast cancer, tamoxifen acts as an estrogen competitive antagonist, and Let-7c enhances the anticancer functions of tamoxifen and reduces the proportion of breast CSCs [78]. The DNA alkylating agent temozolomide (TMZ) is one of the current standard treatment for glioblastoma patients. Dovitinib, a multi-kinase inhibitor, enhances TMZ-induced cytotoxicity through modulation of Lin28/let-7/HMGA2 axis in vitro and in vivo [127].

In NSCLC, combined use of hedgehog inhibitor GDC-0449 leads to sensitization to erlotinib, an EGFR-TKI inhibitor through upregulating let-7c [93]. Let-7 restoration also exerts synergistic effects with radiotherapy to target stem-like cells in breast cancer [81], oral squamous cell carcinoma [98], and melanoma [128]. Both Let-7b [17,84] and let-7d [18,22], are involved in multiple drug sensitization in several preclinical studies, suggesting that tumor delivery of let-7 may be a promising therapeutic approach to enhance drug responses. In head and neck squamous cell carcinoma (HNSCC), the overexpression of let-7a/b miRNAs in combination with CTLA-4 antibody promotes anticancer immunity by accelerating PD-L1 degradation [129].

### 6.3. Let-7 miRNAs Encapsulated in Extracellular Vesicles for Potential Cancer Therapy

Secreted miRNAs encapsulated in extracellular vesicles (EVs) play an important role in cell-cell communications. Exosomes are a subclass of EVs with the size ranging from 30 to 150 nm in diameter containing proteins, RNAs, DNAs, lipids, and metabolites, which mediate juxtacrine/paracrine signaling in the tumor microenvironment [130,131]. Recent studies have indicated a role of exosomal let-7 miRNAs with strong antitumor activities. In non-small-cell lung cancer (NSCLC), let-7e encapsulated in serum exosomes suppresses the metastatic activity of NSCLC cells by inhibiting the SUV39H2/LSD1/CDH1 axis [132].

In pancreatic cancer, exosome secretion from cancer cells promotes the recruitment of pancreatic stellate cells (PSCs) to enhance the formation of distant metastases. This effect is mediated by the transfer of exosomal protein Lin28B, a protein that suppressing let-7 maturation, to PSCs to activate the Lin28B/let-7/HMGA2/PDGFβ signaling cascade [133]. Docosahexaenoic acid (DHA), a long-chain omega-3 polyunsaturated fatty acid with known anticancer properties, promotes exosome secretion and increased exosomal miRNAs, including let-7a, resulting in the inhibition of breast cancer angiogenesis [134].

In a metastatic gastric cancer cell line called AZ-P7a, exosomal release of let-7 miRNAs into the extracellular environment is significantly increased as compared to non-metastatic gastric cancer cells. This increased exosomal transport of let-7 miRNAs is thought to result in depletion of these tumor suppressor miRNAs in the cells, thereby maintaining their invasiveness and metastatic activity [135].

### 6.4. Let-7 Replacement Therapy Using Synthetic Let-7 miRNA Mimics

Use of miRNA mimics is an effective means to restore the normal function of tumor suppressive miRNAs, including let-7 miRNAs. Transfection of chemically synthesized, double-stranded let-7 mimics into the cytoplasm of tumor cells has emerged as a potential strategy for cancer gene therapy. Intratumoral injection of let-7a mimics coupled with lipid vector lipofectamine 2000 has been shown to retard tumor growth in a glioma xenograft model [136].

Similarly, synthetic let-7b mimics intratumorally delivered into H460 lung tumor xenograft model successfully silenced let-7 targeted genes and repress tumor growth [137]. Transfection of Let-7b mimics into gastric tumor cells also attenuate cisplatin resistance and tumor growth by targeting AURKB [138], and HepG2 cells transfected with let-7b mimics display decreased tumorigenic potential through the upregulated expression of p21 [139].

Let-7e expression was found to be significantly reduced in cisplatin-resistant human epithelial ovarian cancer (EOC) cells, and the overexpression of let-7e by transfection of let-7e Agomir, a type of chemically modified double-strand miRNA mimics, reduces the expression of cisplatin-resistance-related proteins, including EZH2 and cyclin D1, and reverses cisplatin resistance when injected locally into EOC xenograft tumors [140].

Despite intensive studies indicating a role of let-7 family of miRNAs as tumor suppressors, the expression of hsa-let-7g (also known as miR-let-7g or let-7g) and its roles in cancer are context-dependent. For example, let-7g exerts anti-tumor effects in gastric cancer and increases the sensitivity of gastric cancer cells to oxidative stress by inhibiting the activation of DNA damage responses [141]. However, in osteosarcoma, patients with higher expression of let-7g displayed a poorer prognosis and lower survival rate, and the overexpression of let-7g promoted the occurrence of osteosarcoma by down-regulating HOXB1 and activating the NFkB pathway [142].

A major challenge in RNA-based cancer therapies is the lack of safe and reliable means for specific delivery of miRNAs to cancer tissues with low cytotoxicity to normal cells. Tremendous progress has been made in restoring let-7 tumor suppressor miRNA function using let-7 miRNA mimics coupled with nanoparticle-based delivery systems, which have been shown to reduce tumor growth successfully in several animal studies [143,144,145].

In a K-Ras-activated autochthonous mouse model of NSCLC, synthetic let-7b mimics conjugated with neural lipid emulsions (NLE) was systemically delivered to lung via intravenous administration, leading to reduced tumor burden [146]. Dendrimer-based polymeric vectors are another form of non-viral vectors with low toxicity and high delivery potency. When administered to a MYC-driven liver tumor model by IV injection, modular degradable dendrimer nanoparticles carrying let-7g miRNA mimic have been shown to inhibit tumor growth and dramatically extended mouse survival [147].

A form of nanocarriers called PEG5k-VE4-DET20 nano-assemblies is also used to deliver let-7b mimic and chemotherapy drug paclitaxel in the non-small cell lung cancer (A549) model through intravenous injection, and paclitaxel and let-7b mimic-loaded nano-assemblies markedly potentiated the cytotoxicity of paclitaxel and retarded tumor growth more efficaciously than Taxol alone [148]. Using an aptamer that binds to and antagonizes the oncogenic receptor tyrosine kinase Axl (GL21.T), Esposito et al. showed that synthesized let-7g miRNA conjugated to the GL21.T aptamer reduced tumor growth in a xenograft model of lung adenocarcinoma [149].

## 7. Concluding Remarks

Let-7, one of the best studied miRNA families, plays critical roles in controlling various biological processes, including self-renewal, proliferation, differentiation, apoptosis and the cellular metabolism through the regulation of a plethora of target genes and signaling pathways. Let-7 miRNAs generally function as tumor suppressors to target multiple oncogenes, including RAS, HMGA2, c-Myc and CCND1. Let-7 miRNAs also function as important immune regulators to control cytokine- and chemokine-mediated inflammatory responses.

As the let-7 family members of miRNAs share a common seed sequence, localizing at nucleotide 2 to 8 in their 5${}^{\prime }$ ends, which is pivotal for target mRNA recognition [8], we predict that individual members of let-7 miRNAs have shared or interchangeable functions in repressing their target genes to inhibit CSC activities.

As a well-recognized signaling node to regulate cancer stemness, Lin28 and let-7 miRNAs form a double negative feedback loop to control tumorigenesis, cancer progression and therapeutic resistance. In more than 15% of tissue malignancies, either Lin28A or Lin28B is reactivated while let-7 is repressed, resulting in the elevated expression of let-7 target genes as well as alterations in tumor cell metabolism.

Current evidence for the Lin28/let-7 axis in regulation of CSC metabolism centers on the glucose metabolism especially aerobic glycolysis, which highlights a role of this axis in the modulation of glycolysis through direct or indirect regulation of PDK1. Although a few studies indicated a role of Lin28/let-7 axis in the regulation of the amino acid and lipid metabolisms, future studies need to decipher the underlying mechanisms by which this axis regulates the metabolism of amino acids and lipids in cancer.

The roles of let-7 miRNAs in the development and diseases, particularly in cancer, have made this family of miRNAs attractive candidates for the development of cancer therapeutics. One of the greatest challenges for RNA-based cancer therapy is to design suitable RNA delivery vehicles to achieve specific and safe delivery of therapeutic miRNAs or miRNA mimics into tumor tissues while avoiding potential toxicities and off-target effects. In this regard, several preclinical nanoparticle-based platforms have been developed to deliver miRNA mimics to tumor tissues with considerable efficacy and low toxicity [150]. For example, MRX34, a miR-34 mimic encapsulated in a lipid carrier (developed by Mirna Therapeutics), has the advantage to become positively charged in the acidic tumor microenvironment, which allows the adherence of MRX34 specifically to tumor cells [151]. In a mouse xenograft model treated with MRX34 nanoparticles by intravenous injection, miR-34 was found to accumulate in tumor tissues, leading to significant tumor regression [150,151]. In 2013, MRX34 was tested in a multicenter phase I trial in patients with various cancers, and a portion of these patients achieved prolonged partial responses or stable disease at the end of trial [150]. However, due to immune-related adverse effects resulting in patient deaths, this trial was prematurely terminated [150]. As recent studies have identified a role of let-7 in regulating CD8 T cell differentiation and function [152,153], future studies in the development of let-7 based RNA therapeutics need to consider the potential immune-related toxicities.

## Figures and Tables

**Figure 1 cells-10-02415-f001:**
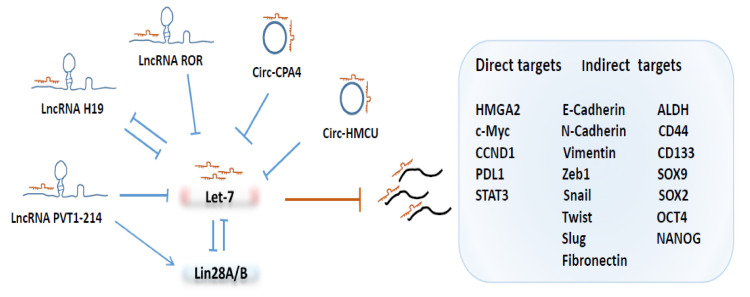
**The regulation of let-7 availability and function by LncRNAs and CircRNAs.** Several lncRNAs and circRNAs act as molecular sponges to antagonize Let-7 function by limiting its access to mRNAs of target genes, such as HMGA2, c-Myc, CCND1, STAT3 and PDL1, leading to indirect regulation of CSC-related factor expression.

**Figure 2 cells-10-02415-f002:**
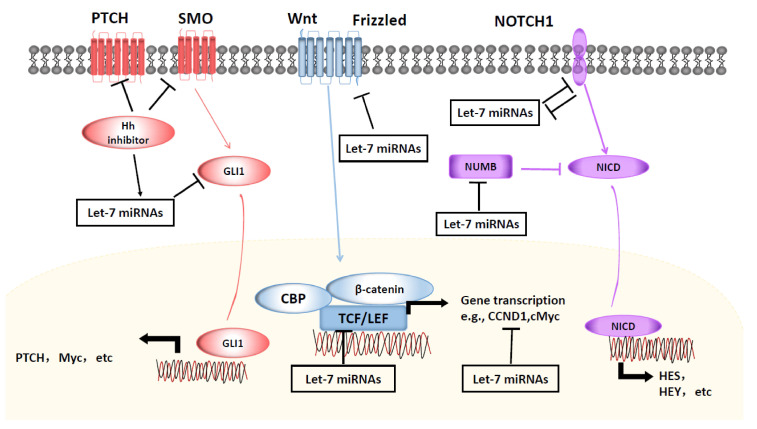
**Let-7 miRNAs regulate CSCs through interaction with the Wnt, Hedgehog and NOTCH pathways.** Members of Let-7 family of miRNAs interact with receptors and/or down-streaming effector genes of Wnt, Hedgehog and NOTCH pathways to regulate CSC self-renewal, proliferation and differentiation.

**Figure 3 cells-10-02415-f003:**
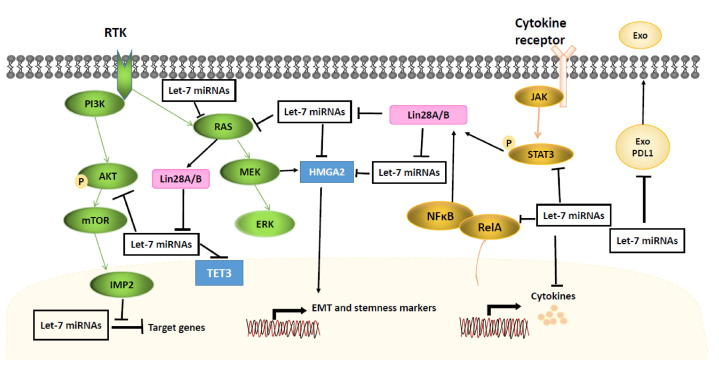
**Let-7 miRNAs regulate CSCs via regulation of STAT3/NFκB/cytokine signaling as well as receptor tyrosine kinases (RTKs)-mediated signaling cascades, including the Ras-RAF-MEK-ERK and PI3K/Akt pathways.** Members of Let-7 family of miRNAs interact with the effector genes of JAK-STAT and NFκB to regulate proinflammatory responses, including cytokine production (**right** part). Let-7 miRNAs, through interaction with down-streaming effectors or target genes of Ras-RAF-MEK-ERK and PI3K/Akt pathways to regulate CSC activities (**left** part).

**Figure 4 cells-10-02415-f004:**
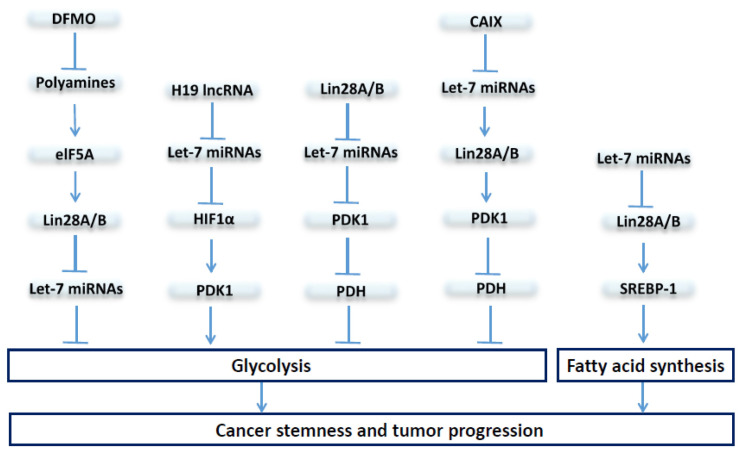
**The let-7/Lin28 axis regulates CSCs through modulation of glucose and fatty acid metabolism.** Lin28A and Lin28B enhance, whereas let-7 suppresses, aerobic glycolysis or fatty acid synthesis mainly through regulation of the PDK1/PDH enzymatic cascade or SREBP-1 to impact CSC activity and tumor progression.

**Table 1 cells-10-02415-t001:** Directly or indirectly repressed genes by the let-7 family of miRNAs in various cancer types.

miRNA	Cancer	Signalling Pathways	Target Genes	Regulated by	Stemness	References
Let-7a	Hepatocarcinoma	Wnt/β-catenin	Direct TCF-4	NA	EMT, sphere formation	[85]
Let-7c	NSCLC	Wnt/β-catenin	Indirect TCF-4 and Wnt1	mir-367,FBXW7	ALDH, CD133, Sphere formation	[86]
Let-7b	NSCLC	Wnt/β-catenin	Indirect CCND1 and TCF-4	Matrine	KLF4, CD133	[83]
Let-7	Osteosarcoma	Wnt/β-catenin	NA	JW74	c-myc	[87]
nano Let-7b	Esophageal cancer	Wnt/β-catenin	Direct TCF-4	NA	ALDH, CD133	[84]
Let-7a, f	Breast cancer	Wnt/β-catenin	Correlation	Lin28B, β-catenin	ALDH	[76]
Let-7c	Breast cancer	Wnt/β-catenin	Direct TCF-4	NA	ALDH, SOX2, NANOG, OCT4	[77]
Let-7c	Breast cancer	Wnt/β-catenin	Direct TCF-4	H19	ALDH	[79]
Let-7d	Breast cancer	Wnt/β-catenin	Direct CCND1	NA	OCT3/4, NANOG, SOX2, ALDH	[81]
Let-7c	Breast cancer	Wnt/β-catenin	Direct ERa	NA	ALDH	[78]
Let-7c	Breast cancer	Wnt/β-catenin	Direct TCF-4	mir-146a, Lin28A, H19	ALDH	[80]
Let-7b	Breast cancer	Wnt/β-catenin	Indirect CCND1, TCF-4	Matrine, Lin28A	CD133, KLF4	[82]
Let-7	Breast cancer	STAT3/NFκB	Indirect HMGA2	M1	ALDH, CD44+/CD24-, KLF4, NANOG	[96]
Let-7c	Keratinocytes	STAT3/NFκB	Indirect k-Ras, p-RelA	Arsenite	K5, CD34	[97]
Let-7	NSCLC	STAT3/NFκB	Indirect HMGA2, TGFBR3	MUC1-C, NFκB, Lin28b	Sphere formation	[94]
Let-7c	Oral cancer	STAT3/NFκB	Direct IL8	NA	ALDH, CD44+	[98]
Let-7	Breast cancer	Metabolism	Direct HIF-1α, indirect PDK1	H19	ALDH, OCT4	[106]
Let-7	Hepatocarcinoma	Metabolism	Indirect SREBP-1	Lin28A/B	Fatty acid synthesis, tumor growth	[107]
Let-7	Neuroblastoma	Metabolism	NA	DMFO, Lin28B, MYCN	Neurosphere formation	[108,109]
Let-7c, d	Breast cancer	Metabolism	Indirect PDK1, p-PDH	CAIX, Lin28B, NFκB	ALDH, NANOG	[110]
Let-7b, g	Glioblastoma	MAPK/PI3K	Correlation p-MAPK	Lin28A, K-Ras	CD133, SOX2, Nestin, OLIG2, OCT4, Snail	[101]
Let-7a, f, g	NSCLC	MAPK/PI3K	Indirect pAKT, cMYC, VEGFA	Lin28B	CD44+/CD326+, EMT	[105]
Let-7a	Pancreatic cancer	MAPK/PI3K	HMGA2	MEK	EMT	[104]
Let-7i	Pancreatic cancer	MAPK/PI3K	Direct TET3	K-Ras→Lin28B	OCT4, NANOG, SOX2, CD133	[102]
Let-7a	Pancreatic cancer	MAPK/PI3K	K-Ras	Green tea catechins (GTC)	ALDH, sphere formation	[103]
Let-7	Pancreatic cancer	Hedgehog	Correlation Shh	NA	CD133, CXCR4, sphere formation	[92]
Let-7c	NSCLC	Hedgehog	NA	Hedgehog inhibitor	SOX2, NANOG, EpCAM	[93]
Let-7b	Breast cancer	NOTCH	Direct NUMB	mir-129→ESR1→DICER1	CD44+/CD24-, ALDH	[91]
Let-7a	Breast cancer	EMT	Indirect ERa, pS2, CCND1	NA	ALDH, SP population	[21]
Let-7d	Oral cancer	EMT	Indirect Twist, Snail, CDH1, Fibronectin 1	NA	ALDH	[22]
Let-7	Ovarian cancer	EMT	NA	Snail	CD117, CD133, CD44, sphere formation	[24]
Let-7	Ovarian cancer	EMT	Correlation	Correlation	CD44, CD117, CD133, OCT4, NANOG, Lin28A, HMGA2	[25]
Let-7	Pancreatic cancer	EMT	Correlation	LncRNA ROR	CD133, CD44, ALDH	[72]
Let-7a	Tongue squamous cell carcinoma	EMT	Direct HMGA2	H19	CDH1, CDH2, Vimentin, Snail, Zeb1, Twist1	[23]
Let-7b	Gastric cancer	Pluripotency transcription factor	Direct c-myc	NA	c-myc	[17]
Let-7	Prostate cancer	Pluripotency transcription factor	Direct HMGA2	Lin28B	SOX2, CD44, CD133	[19]
Let-7	Bladder cancer	Pluripotency transcription factor	Indirect SOX2	COX2, PGE2	SOX2, OCT4, NANOG, CD24, CD133	[18]
Let-7	Pancreatic cancer	Pluripotency transcription factor	Correlation	Correlation	CD44+ /CD133+ /EpCAM+ cells, Snail, OCT4, NANOG, SOX2, EZH2	[16]

**Table 2 cells-10-02415-t002:** Induction of Let-7 miRNAs to overcome therapeutic resistance by targeting CSCs.

Therapy	miRNAs	Regulated Stemness Markers	Functions of miRNAs in Therapeutic Sensitization	References
Mutiple drugs	Let-7	CD133, CD24	COX2 inhibitor→Let-7→SOX2→CD133, CD44	[18]
	Let-7d	ALDH	Let-7d→EMT→ALDH	[22]
	Nano Let-7b	CD133, ALDH	Nano Let-7b→TCF4/Wnt→CD133, ALDH	[84]
	Let-7b	EMT	Let-7b→cmyc→EMT	[17]
Temozolomide	Let-7	Sphere formation	Dovitinib→Let-7/Lin28a/HMGA2→pSTAT3→Sphere formation	[127]
Tamoxifen	Let-7c	ALDH	Let-7c→TCF4/Wnt→ALDH	[78]
Erlotinib/Cisplatin	Let-7	SOX2, NANOG, EpCAM	Hedgehog inhibotor→Let-7→EMT→SOX2, NANOG, EpCAM	[93]
Cisplatin	Let-7b	CD133	Matrine→Let-7b/Lin28A→CCDN1/Wnt→CD33	[82]
	Let-7	CD133, CD44, CD117	shSnail/HGSOC→Let-7→CD133, CD44, CD117	[24]
Cisplatin/Radiation	Let-7c	ALDH, CD44	Let-7c→IL8→ALDH, CD44	[98]
Radiation	Let-7d	ALDH	Let-7d→CCND1/Wnt→ALDH	[81]
	Let-7	Sphere formation	Let-7/Lin28B→γH2AX	[128]
Gemcitabine	miR-98	CD133	Thiostrepton→miR-98→CD133	[126]
	Let-7	CD133	Hedgehog inhibitor→Let-7→CD133	[92,125]
CTLA-4 antibody	Let-7a/b	Wnt/β-catenin	Let-7a/b→TCF-4→β-catenin/STT3→PDL1 degradation	[129]

## Data Availability

Not applicable.

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
