# Peer review of "The Roles of the Let-7 Family of MicroRNAs in the Regulation of Cancer Stemness"

_cells, 2021, doi:10.3390/cells10092415_

Round 1
Reviewer 1 Report
The authors wrote a comprehensive overview of the role of let-7 family in CSC biology including how let-7a being regulated. Some minor revisions are suggested.
- Figure 1: It's better to be consistent that using gene name or common name for the genes being indicated.
- It is suggested to make a summary table that presents the direct/indirect targets of the members of let-7 family in each different cancer type.
- Some evidence indicates that let-7s could be delivered by exosomes and exosomes are considered as a promising biological nanoparticle for drug delivery. To add a subsection of the exosome issue will strengthen the manuscript that attracts more readers to cite.
Author Response
We would like to express our gratitude to reviewer 1 who critically read our manuscript and made constructive comments to improve the manuscript. We have modified Figure 1 as author suggested and made a summary table (Table 1) to present the direct or indirect target genes for each of let-7 members of miRNAs. We also add a subsection in 6.3. Let-7 miRNAs encapsulated in extracellular vesicles for potential cancer therapy, to summarize current evidence of using let-7 miRNAs encapsulated in exosomes as a promising therapeutic strategy.
Reviewer 2 Report
Authors: Yuxi Ma, Na Shen, Max Wicha, Ming Luo
Title: The Let-7 Family of MicroRNAs as Master Regulators of Cancer Stemness
Comments and suggestions:
The authors have submitted a comprehensive review dedicated to the regulatory role of Let-7 family of miRNAs in the development and maintenance of cancer stemness. I think that this review article will be interesting for readers of the journal Cells. There are, though, a couple of critical remarks:
1) I would recommend to cite and discuss the relevant reference doi: 10.1172/JCI89212 (Lai et al. J Clin Invest 2018, 128(2):625-643). In that paper, the relevant mechanism is described which comprises Dicer, Let-7 family of miRNAs and HIF-1alpha, while the latter (HIF-1alpha) is known to be one of the drivers of EMT and generation of CSC-like cells occurring in hypoxic niches (you may also cite a ref: doi: 10.3390/cancers13051102).
2) The design of this manuscript does not meet the template given in the instructions for authors of the journal Cells. The authors should prepare the next version better corresponding the template.
Author Response
We would like to express our gratitude to the reviewer who critically read our manuscript and made constructive comments to improve the manuscript. As the reviewer recommended, we have cited and discussed the relevant reference doi: 10.1172/JCI89212 [Lai et al. J Clin Invest 2018, 128(2):625-643], which is in line 137-142 at the end of the first paragraph in section 3.2. Post-transcriptional regulation of let-7 by RNA-binding/processing proteins. We also edited the original manuscript using the template software download from the journal Cells.
Reviewer 3 Report
In their manuscript entitled ‘The Let-7 Family of MicroRNAs as Master Regulators of Cancer Stemness’, Ma and colleagues summarized the suppressive roles of let-7 family microRNAs in cancer stem cells (CSCs) of solid tumors. The authors described the biogenesis and regulation of let-7 family microRNAs (miRNAs) as well as their roles in the signaling pathways involved in CSC maintenance. Finally, the authors present the possible application of synthetic let-7 miRNAs in cancer therapies.
While the authors systematically summarize various aspects of tumor suppressor let-7 miRNAs in CSCs, the Figures are less organized to convey the contents of this review. It is also unclear whether the authors consider that each member of let-7 family miRNAs has distinct functions, or it is functionally interchangeable.
The concerns the reviewer feels are listed below.
Major points:
- Considering many miRNAs are involved in the regulation of CSCs (i.e. ‘Role of miRNA-Regulated Cancer Stem Cells in the Pathogenesis of Human Malignancies.’ Cells. 2019; 8(8): 840. PMID: 31530793), I think that the title of this review may overstate the roles of let-7 miRNAs in CSCs.
- It is unclear whether the authors consider each member of let-7 miRNAs is functionally interchangeable or not. I would suggest describing the fact that the seed sequences of let-7 miRNAs are shared and therefore it is considered that they are functionally interchangeable with each other. I would also suggest using ‘let-7 family miRNAs’ in the Figures, instead of presenting the names of each let-7 family miRNA. For example, in Figure 2, the specific family member ‘let-7c’ is presented to target GLI1 and Frizzled, suggesting that no other family member of let-7 miRNAs does not have an ability to target them. Similar issues are found in Figure 3 and Figure 5.
- I would suggest strengthening the description of the post-transcriptional control of let-7 miRNA biogenesis by including the terminal uridylation of let-7 by TUTase (please refer to ‘Post-transcriptional control of miRNA biogenesis.’ RNA 2019. 25: 1-16. DOI, 10.1261/rna.068692.118).
- The contents of Figure 5 will be better to be presented as a Table.
- In Figure 4, ‘let-7-Lin28’ interaction is presented 4 times, making this figure difficult to be understood at a glance.
Minor points:
- I would suggest describing the similarities and differences among let-7 family miRNAs more in detail using Figures.
- Please edit the sentences starting by Although accumulating evidence indicating… between line 253 and 258; and those starting by The activation of transcription factors… between line 268 and 273.
- I think that PDK1 reduces the activity of PDH and suppresses aerobic glycolysis. If let-7 targets PDK1 (Figure 4), it will enhance aerobic glycolysis. This notion is contrary to the description in line 332.
Author Response
We would like to express our gratitude to the reviewer who critically read our manuscript and made constructive comments to improve the manuscript. We have addressed the major and minor points raised by the reviewer as indicated as follows:
Major point 1, “Considering many miRNAs are involved in the regulation of CSCs (i.e. ‘Role of miRNA-Regulated Cancer Stem Cells in the Pathogenesis of Human Malignancies.’ Cells. 2019; 8(8): 840. PMID: 31530793), I think that the title of this review may overstate the roles of let-7 miRNAs in CSCs”.
We agree with the reviewer and have modified the title of the manuscript as “The Roles of Let-7 Family of MicroRNAs in Regulation of Cancer Stemness”.
Major point 2: “It is unclear whether the authors consider each member of let-7 miRNAs is functionally interchangeable or not. I would suggest describing the fact that the seed sequences of let-7 miRNAs are shared and therefore it is considered that they are functionally interchangeable with each other. I would also suggest using ‘let-7 family miRNAs’ in the Figures, instead of presenting the names of each let-7 family miRNA. For example, in Figure 2, the specific family member ‘let-7c’ is presented to target GLI1 and Frizzled, suggesting that no other family member of let-7 miRNAs does not have an ability to target them. Similar issues are found in Figure 3 and Figure 5.”
We agree with the reviewer and have added a sentence in the concluding remarks in line 511-514 to reflect reviewer’s point that functions of let-7 miRNAs may be overlapped and interchangeable, because the let-7 family members of miRNAs share a common seed sequence, localizing at nucleotide 2 to 8 in their 5′ ends that is pivotal for target mRNA recognition. We also modified all figures using let-7 or let-7 miRNAs instead of let-7a, let-7c, and others.
Major point 3: “I would suggest strengthening the description of the post-transcriptional control of let-7 miRNA biogenesis by including the terminal uridylation of let-7 by TUTase (please refer to ‘Post-transcriptional control of miRNA biogenesis.’ RNA 2019. 25: 1-16. DOI, 10.1261/rna.068692.118).”
We agree with the reviewer and have cited and discussed the relevant reference in line 151-155, section 3.2. Post-transcriptional regulation of let-7 by RNA-binding/processing proteins.
Major point 4: “The contents of Figure 5 will be better to be presented as a Table.”
We agree with the reviewer and have replaced Figure 5 with Table 2.
Major point 5: “In Figure 4, ‘let-7-Lin28’ interaction is presented 4 times, making this figure difficult to be understood at a glance.”
We agree with the reviewer and have modified Figure 4 accordingly.
Minor point 1: “I would suggest describing the similarities and differences among let-7 family miRNAs more in detail using Figures.”
We agree with the reviewer and have added a table to list the direct or indirect target genes of each let-7 family member of miRNAs in various cancer types with references. We also added a sentence in the concluding remarks (see line 511-514) to reflect reviewer’s point that the functions for each member of let-7 miRNAs may be overlapped and interchangeable, because let-7 family members of miRNAs share a common seed sequence that is pivotal for target mRNA recognition.
Minor point 2: “Please edit the sentences starting by Although accumulating evidence indicating… between line 253 and 258; and those starting by The activation of transcription factors… between line 268 and 273.”
As the reviewer suggested, we have edited the corresponding sentences.
Minor point 3: “I think that PDK1 reduces the activity of PDH and suppresses aerobic glycolysis. If let-7 targets PDK1 (Figure 4), it will enhance aerobic glycolysis. This notion is contrary to the description in line 332.”
We would like to clarify that the function of PDK1 is to phosphorylate PDH to inhibit PDH enzymatic activity. As PDH is the key enzyme to direct glucose influx to mitochondrial oxidative phosphorylation (OXPHOS), inhibited PDH enzymatic activity will redirect glucose flux from entering TCA cycle to aerobic glycolysis and lactate production. Therefore, let-7 by targeting PDK1 will enhance PDH activity, promoting OXPHOS but inhibiting aerobic glycolysis. We have added a sentence in line 342-345 of section 5, The let-7/Lin28 axis in regulation of stem cell metabolism, to reflect this point.
Round 2
Reviewer 3 Report
Ma and colleagues properly responded to my concerns and comments and now I found significant improvements in the manuscript. The revised manuscript is attractive enough for researchers interested in the roles of let-7 miRNAs and cancers. I think that the manuscript is acceptable for publication in Cells.
I have one comment for consideration during the proof-reading process.
- Although I continue to think that let-7 promotes anaerobic glycolysis (that produce lactate), it is up to the authors how describe this phenomenon (line 342-345, authors’ response to Minor point 3).